# Plasma 25(OH)D Concentrations and Gestational Diabetes Mellitus among Pregnant Women in Taiwan

**DOI:** 10.3390/nu13082538

**Published:** 2021-07-25

**Authors:** Thu T. M. Pham, Ya-Li Huang, Jane C.-J. Chao, Jung-Su Chang, Yi-Chun Chen, Fan-Fen Wang, Chyi-Huey Bai

**Affiliations:** 1School of Public Health, College of Public Health, Taipei Medical University, Taipei 110-31, Taiwan; phamminhthu.ytcc@gmail.com; 2Faculty of Public Health, Hai Phong University of Medicine and Pharmacy, Hai Phong 042-12, Vietnam; 3Department of Public Health, School of Medicine, College of Medicine, Taipei Medical University, Taipei 110-31, Taiwan; ylhuang@tmu.edu.tw; 4School of Nutrition and Health Sciences, Taipei Medical University, Taipei 110-31, Taiwan; chenjui@tmu.edu.tw (J.C.-J.C.); susanchang@tmu.edu.tw (J.-S.C.); yichun@tmu.edu.tw (Y.-C.C.); 5Nutrition Research Center, Taipei Medical University Hospital, Taipei 110-31, Taiwan; 6Master Program in Global Health and Development, College of Public Health, Taipei Medical University, Taipei 110-31, Taiwan; 7Graduate Institute of Metabolism and Obesity Sciences, College of Nutrition, Taipei Medical University, Taipei 110-31, Taiwan; 8Chinese Taipei Society for the Study of Obesity (CTSSO), Taipei 110-31, Taiwan; 9Master Program in Food Safety, College of Nutrition, Taipei Medical University, Taipei 110-31, Taiwan; 10Division of Endocrinology and Metabolism, Department of Medicine, Taipei Veterans General Hospital, Taipei 110-31, Taiwan; doc1298d@yahoo.com.tw; 11Department of Medicine, Yangming Branch, Taipei City Hospital, Taipei 111-31, Taiwan

**Keywords:** gestational diabetes mellitus, 25(OH)D concentration, vitamin D deficiency, pregnant women

## Abstract

Vitamin D’s function in the development of gestational diabetes mellitus (GDM) is not consistent in the literature. We examined the association between maternal plasma 25(OH)D concentration and GDM risk. A national cross-sectional study (1497 pregnant women) was conducted between 2017 and 2019 across Taiwan. Blood samples were drawn at recruitment to assess 25(OH)D concentrations, including vitamin D deficiency (VDD) (<20 ng/mL), insufficiency (<32 ng/mL), and sufficiency (≥32 ng/mL). GDM was detected from 24 to 28 weeks of gestation with the results extracted from the antenatal visit records. The prevalence of GDM was 2.9%. Logistic model analysis showed that 25(OH)D concentrations were not significantly associated with the risk of GDM (adjusted odds ratio (AOR) = 0.97, *p* = 0.144). However, subjects with VDD had a significantly greater risk of GDM (AOR = 2.26, *p* = 0.041), but not in those with vitamin D insufficiency (AOR = 1.20, *p* = 0.655). Furthermore, cubic piecewise spline regression was used to explore the relationship between five-unit intervals of 25(OH)D and the predicted probability of GDM. As the proportion of GDM increased for low 25(OH)D concentrations, it decreased at moderate concentrations and increased again at higher concentrations. These findings revealed a nonlinear relationship between 25(OH)D and GDM risk. VDD would be risky for GDM occurrence.

## 1. Introduction

Gestational diabetes mellitus (GDM) is a type of diabetes characterized by glucose intolerance, which has its onset or is first detected during pregnancy [1]. GDM is reported to have an increasing prevalence [2,3]. It has emerged as a global public health concern because it affects short- and long-term maternal and neonatal health and places a significant financial burden on the healthcare services [4,5,6]. Although exploring the mechanisms leading to GDM is an optimal way to improve its prevention and treatment, it remains a challenge. Some studies have actively investigated the causes of GDM, such as interest in insulin resistance, as a major cause [7]. Additionally, low vitamin D level has been identified as a possible cause in a growing number of studies.

Vitamin D has an essential role in pregnancy because of its impact on many aspects such as bone health, calcium homeostasis, immunomodulation between mother and fetus, implantation/placentation, respiratory maturation, and the prevention of pre-eclampsia [8,9]. However, the proportion of vitamin inadequacy has increased significantly in women during pregnancy [10] and reproductive age [11], which may lead to adverse pregnancy outcomes. In the literature review, vitamin D deficiency (VDD) during pregnancy resulted in hypertensive disorders such as preeclampsia and GDM, fetal growth restriction, small gestational age, preterm labor, and neonatal death [12,13]. Studies have examined the relationship between VDD and the development of GDM, as VDD is related to insulin resistance and epigenetic modifications, which are features of diabetes [14].

Reports on the relationship between vitamin D and GDM have been contradictory in the existing literature. Although the latest meta-analysis of observational studies revealed that 25-hydroxyvitamin D (25(OH)D) concentration had a significantly converse relationship with the GDM development [15,16,17,18], other studies indicated non-significant results on this association [19,20]. Moreover, vitamin D significantly impacts GDM occurrence after reaching a certain level, but the relationship between 25(OH)D and GDM is nonlinear [21,22]. Many authors have provided conflicting results, and the data regarding vitamin D levels associated with GDM are unclear and exceptionally scarce in Taiwan. Therefore, we aimed to explore the relationship between plasma 25(OH)D concentrations and the risk of GDM among pregnant women in Taiwan.

## 2. Materials and Methods

### 2.1. Study Population

The data were obtained from the Nutritional Survey of Pregnant women in Taiwan (NPWT). The survey included a cross-sectional study from June 2017 to February 2019, which was conducted on all pregnant women aged ≥ 15 years residing in Taiwan, including indigenous residents or those being approved with an alien resident certificate. The study population was selected based on a multiple-stage sampling plan. First, eight layers were separated by geographical location (north, center, south, and east of Taiwan) and hospital size (large and small) based on the number of women availing pregnancy-related services per year. The prenatal inspection includes 10 routine and three special inspections (ultrasonic screening). The routine includes medical history inquiry, height and weight measurement, blood pressure, basic physical examination, urine test, fetal heart sound monitoring, glucose, complete blood count, blood type, hepatitis B, HIV (Human Immunodeficiency Virus), syphilis, and German measles. The prenatal inspection from governments covers over 99% of pregnant women in Taiwan. Second, the sampling probability of hospital was set based on the probability proportional to size in each layer. The total serving amount of hospital was conducted from the list of receiving service certifications in the National Health Insurance program of the government (2014–2016). Eleven hospitals (clusters) were randomly sampled from a list of hospitals. A total of 150 to 300 participants from one or two hospitals in each layer were enrolled based on the potential size of annual outpatients. All pregnant women arriving in these hospitals and those who underwent antenatal examinations were recruited during the study period. The response rate of each hospital equals the number of enrolled cases per day divided by the number of inspection outpatients, with approximately 30–40% on weekdays and 60–80% on weekends.

The sample size of 1062 was calculated based on the annual population of 200,000 women with new birth within the study period, 95% confidence interval, and 3% margin of error. In fact, we enrolled 1502 pregnant women in the Nutritional Survey after excluding those who had non-singleton pregnancies, could not understand and speak Mandarin, and could not complete the questionnaires. All the participants were asked to obtain informed consent.

### 2.2. Data Collection

The standardized questionnaires, food frequency questionnaires and 24 h dietary records were validated by experts. All records were administered by well-trained interviewers. The information of prenatal inspection was recorded from first inspection to baby birth. Blood and urine of studied subjects were also collected.

#### 2.2.1. Assessment of GDM

The GDM status was extracted from the prenatal visit records of participants. Obstetricians diagnosed women with GDM via blood tests after ingestion of 75 g glucose during 24–28 weeks of pregnancy. The one-step diagnosis procedure was applied according to the World Health Organization criteria such that either fasting plasma glucose (FPG) ≥ 92 mg/dL (5.1 mmol/L), one-hour plasma glucose (PG) ≥ 180 mg/dL (10.0 mmol/L), or two-hour PG ≥ 153 mg/dL (8.5 mmol/L) [1].

#### 2.2.2. Assessment of Vitamin D Levels

Blood samples were drawn at the time of recruitment when the average gestational age was 29.7 weeks, then froze at −80 °C, and analyzed in batches. An electrochemiluminescence immunoassay (ECLIA) was used to quantify the total plasma concentration of 25(OH)D. The concentration was measured in duplicate and averaged of both D2 and D3. The total coefficient of variation (CV) for 25(OH)D was 5–7%. The reference range was from 20 to 100 ng/mL. The classification of vitamin D levels was based on the recommendation of the Health Promotion Administration, Ministry of Health and Welfare in Taiwan, including VDD if its level was less than 20 ng/mL, insufficiency if the level was under 32 ng/mL, and sufficiency if it was at least 32 ng/mL.

#### 2.2.3. Assessment of Covariates

Socio-demographic factors and pregnancy-related data were obtained from all participants at enrollment using self-reported questionnaires, including maternal age (years), education level, living area, monthly household income, religion, body height (cm) and weight (kg) before pregnancy, current weight, gestational age, parity, gravidity, and personal history of diabetes. The maternal pre-pregnancy body mass index (BMI, kg/m^2^) was calculated, and gestational weight gain at recruitment was estimated by subtracting the pre-pregnancy weight from the current weight. Pregnant women reported their frequency of using vitamin D and/or calcium supplements during pregnancy as “never use,” “use less than one day per week,” “use 2–5 days/week,” and “use almost daily.” Then, this factor was recoded into two categories of usage, yes or no, due to the small sample size.

The 24-h dietary intake was recorded to assess the intake of total energy (kcal), raw protein (g), raw fat (g), total carbohydrates (g), vitamin D content (mg), and the use of vitamin supplements. The percentages of calories from protein, fat, and carbohydrates were calculated using the Formulas (1)–(3):(1)$$\%\mathrm{Protein}=\frac{\mathrm{raw}\;\mathrm{protein}\times 4\times 100}{\sum^{\;}\mathrm{calories}}$$
(2)$$\%\mathrm{Fat}=\frac{\mathrm{raw}\;\mathrm{fat}\times 9\times 100}{\sum^{\;}\mathrm{calories}}$$
(3)$$\%\mathrm{Carbohydrate}=\frac{\mathrm{total}\;\mathrm{carbohydrates}\times 4\times 100}{\sum^{\;}\mathrm{calories}}$$

The dosages of supplements were calculated if participants provided the exact dosage. However, these parameters were missing a lot, especially the brands and models of vitamins. Therefore, in the current study, we only analysed the frequency of usage of vitamin D-only or D-based supplements.

As vitamin D levels were influenced mainly by sunlight [23,24], a number of confounders with 25(OH)D levels related to sunlight were assessed. The sun exposure was estimated through the question “did you expose to the outdoor sunlight last month?”, and the answer included two categories of “no” exposure if less than 10 min per day, and “yes” if exposed for more than 10 min per day. The sample collection dates were categorized into the season of blood draw, including sunny months from June to November and rainy months from December to May.

### 2.3. Ethical Consideration

This study was funded by the Health Promotion Administration, Ministry of Health and Welfare in Taiwan (C1050912), and approved by the Institutional Review Board of the government and selected hospitals (IRB number: N201707039).

### 2.4. Statistical Analysis

The sampling weight was calculated to account for the non-response and cluster sampling. Goodness tests were used to check the distributions between the sample and population. In this study, all analyses included a sample-weighting approach.

First, we performed a descriptive analysis to explore the distribution of the different variables. To compare the distribution of studied variables between women with and without GDM, we used chi-square tests and *t*-tests (or Mann–Whitney test) for categorical and continuous variables, respectively. Second, we utilized logistic regression analysis to investigate the association between 25(OH)D levels and GDM. In regression analysis, gestational age and supplement usage were regrouped, and personal history of diabetes was removed because of the small sample size. Two models were built. Model 1 was adjusted for factors associated with GDM (*p* < 0.05) in univariate analysis, including maternal age, monthly household income, religion, pre-pregnancy BMI, gestational weight gain, gestational age, %protein, %carbohydrate, vitamin D, and/or calcium supplement. Model 2 was further adjusted for sun exposure and the season of blood draw. The coefficients of <0.3 were tested using Spearman correlation to avoid multicollinearity in the multivariate models. The analysis was performed using 25(OH)D concentrations as continuous and categorical variables. The categories of 25(OH)D included VDD (<20 ng/mL vs. ≥20 ng/mL) and vitamin D insufficiency (<32 ng/mL vs. ≥32 ng/mL). Odds ratios (ORs) and 95% confidence intervals (95% CIs) were reported, and the significant level was considered at *p*-values < 0.05.

We further calculated the predicted probability of GDM (and 95% CI) using a logistic model adjusted for all covariates in Models 1 and 2. Then, cubic piecewise spline regression was used to express the relationship between 5-unit intervals of 25(OH)D and the predicted probability of GDM.

All analyses were performed using IBM SPSS v22.0 (IBM Corp., Armonk, NY, USA) and SAS (v9.4; SAS, Chicago, IL, USA).

## 3. Results

### 3.1. Characteristics of Study Participants

As shown in Table 1, among the 1497 pregnant women, the number of women with GDM accounted for 2.9%. The mean 25(OH)D concentration was 25.5 ± 8.9 ng/mL. Compared to participants without GDM, those with GDM had a higher age (*p* = 0.038), higher pre-pregnancy BMI (*p* = 0.007), higher gestational weight gain (*p* < 0.001), higher gestational age (*p* < 0.001), higher percentage of history of diabetes (*p* = 0.007), and consumed more vitamin D and/or calcium supplements (*p* = 0.012). The proportion of women with GDM varied according to the different categories of religion (*p* = 0.001). In addition, the percentages of protein and carbohydrate intakes were different between women with GDM and without GDM, with *p* = 0.037 and 0.017, respectively.

### 3.2. Levels of Plasma Vitamin D during Pregnancy and Association with GDM

The proportions of VDD and vitamin D insufficiency were 30.3% and 47.7%, respectively. During pregnancy, vitamin D levels varied markedly, with the VDD proportion being highest in the first trimester and lowest in the third trimester, respectively (*p* < 0.001). The mean 25(OH)D levels among women with and without GDM were 26.6 ng/mL and 25.5 ng/mL, respectively (*p* = 0.426). In addition, VDD between women with GDM (2.9%) and without GDM (3.0%) was not significantly different with *p* = 0.903.

The crude ORs of GDM risk in terms of socio-demographics, pregnancy-related characteristics, dietary intakes, and supplements are represented in Table 2, where factors associated with GDM at *p* < 0.05 were selected for adjustment in the multivariate analysis. In the multivariate analysis (Table 3), plasma 25(OH)D concentration was not significantly associated with GDM risk (adjusted OR = 0.97, *p* = 0.144). However, the likelihood of GDM was significantly higher in pregnant women with VDD (levels <20 ng/mL; AOR = 2.26, *p* = 0.041) than for those with vitamin D insufficiency (AOR = 1.20, *p* = 0.655). Owing to this inconsistency, the adjusted GDM risk was further plotted on five-unit intervals of 25(OH)D concentrations (see Figure 1). The results indicated that the occurrence of GDM was not constant with increasing 25(OH)D concentrations, revealing a nonlinear relationship between 25(OH)D and the GDM risk. Several peaks were observed following the 25(OH)D concentration. A slightly upward trend was observed at approximately 27 ng/mL, followed by a downward trend from 27 to 35 ng/mL. The GDM risk trend increased again and continued to increase at >35 ng/mL of 25(OH)D concentrations. The 95% CIs were stable for 25(OH)D levels ranging from 10 to 50 ng/mL. This means that after adjusting all covariates, the GDM risk is significantly higher at 25(OH)D levels <20 ng/mL (Table 3 and Figure 1) and >35 ng/mL (Figure 1).

## 4. Discussion

Thus far, the current study is the first to assess the link between plasma 25(OH)D concentration during pregnancy and the risk of GDM in Taiwan. In this study, we found that this association was nonlinear. Although VDD was significantly associated with an increased risk of GDM, the risk did not decrease as vitamin D levels increased.

Reports concerning the association between 25(OH)D concentration and GDM across studies were contradictory [25]. A unique cohort study in China showed a positive association between a high level of 25(OH)D (≥30 ng/mL) and developing GDM [26]. Several studies have found a non-significant association between low vitamin D levels and the risk of GDM [27,28,29]. Others have shown a significant association between VDD (<20 ng/mL) and a greater risk of GDM. For instance, a 2018 meta-analysis of 29 articles [30], a 2019 meta-analysis of 36 articles [31], and a 2020 meta-analysis of 21 articles [32] showed that VDD or a low level of vitamin D accounted for an elevated risk of GDM. Similarly, Xia et al. found that VDD in the first trimester was risky for developing GDM among US pregnant women, and the risk of GDM increased as VDD persisted through the second trimester [33]. Likewise, several cohort studies conducted in India have indicated that a low level of plasma 25(OH)D in early pregnancy contributed to a high risk of developing GDM [34,35].

Our results were in line with studies reporting that VDD increases the risk of GDM. However, it was interesting to note that the relationship between 25(OH)D levels and GDM risk was not linear because GDM risk was high for low levels, lower for moderate levels, and, finally, increased for higher levels of 25(OH)D. Our findings were supported by Salakos et al. [21] that GDM risk increased with vitamin D levels up to 25 ng/mL, then decreased as vitamin D levels ranged from 25 to 40 ng/mL, and, finally, increased for higher levels of vitamin D. The strength of the association between VDD and GDM possibly depends on the internal variation of individuals. For example, 25(OH)D levels < 50 nmol/L did not affect GDM risk, and a reduction of GDM risk was observed only when pregnant women had a mean 25(OH)D level greater than 50 nmol/L and regularly used 400–600 IU vitamin D supplement [22]. For pregnant women with VDD, obese people are more likely to develop GDM than those with normal weight [36].

The pathogenesis of GDM caused by VDD is unclear, but the attendance of vitamin D in causing GDM risk has been demonstrated in various studies. On the one hand, a link between vitamin D and insulin secretion was mentioned. From the molecular perspective, vitamin D regulates Ca^2+^ homeostasis, which modulates insulin secretion [37]. From an individual perspective, Sedighi et al. [38], via a randomized controlled trial, reported that insulin resistance in women with GDM was reduced significantly by the use of vitamin D supplements. However, other studies have reported that no relationship between VDD and insulin resistance existed [39]. On the other hand, some vitamin D-related genes modified vitamin D levels underlying GDM pathogenesis. For instance, Wang et al. [40] reported that vitamin D concentrations were low and varied according to the expression levels of genes related to vitamin D among subjects with GDM compared to those without GDM.

Our study was strong in assessing many factors accounting for the variability in plasma 25(OH)D concentration, such as sunlight exposure, season, vitamin supplementation, and dietary intake. The adjustment for a large number of potentially confounding risk factors allowed us to estimate the independent effect of VDD on GDM. Second, we used the sampling weights method to ensure a representative sample for the whole population. However, several limitations of our study should be considered. First, several parameters influencing vitamin D levels and GDM risk (e.g., physical activity, smoking) were not assessed. Second, plasma insulin and hemoglobin A_1_c (HbA_1_c) levels were not estimated. Third, the causal relationship between VDD and GDM could not be interpreted in a cross-sectional study. Lastly, we did not include data related to brand, dosage of vitamin D and calcium supplements in the current analysis, and the data related to vitamin D and calcium supplements were self-reported, which may have resulted in under- or over-reporting bias.

## 5. Conclusions

Our findings revealed a nonlinear relationship between plasma 25(OH)D concentration and the risk of GDM. Our results highlight the possible role of vitamin D deficiency in developing GDM, which may help with developing appropriate interventions among pregnant women. However, sufficient vitamin D levels that could protect pregnant women from GDM risk were not established in this study. We suggest that further studies, including well-designed randomized controlled trials, are required to confirm our findings and explore the potential effects of vitamin D supplements on preventing GDM risk.

## Figures and Tables

**Figure 1 nutrients-13-02538-f001:**
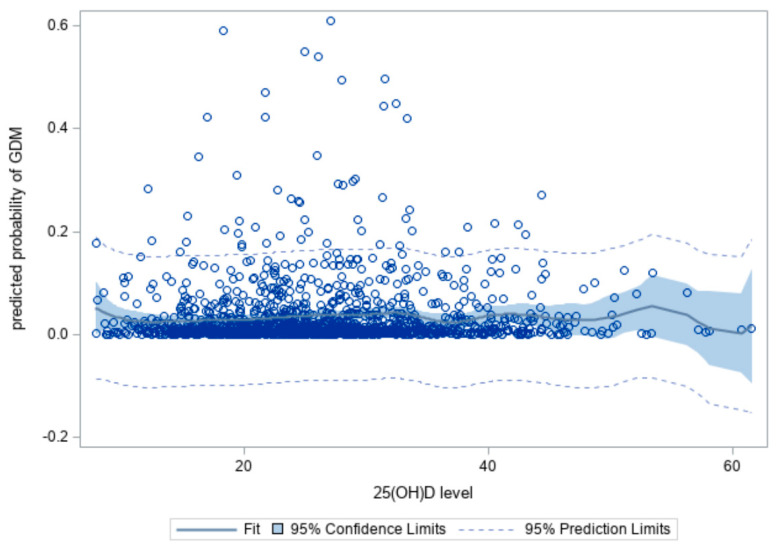
Association between five-unit intervals of plasma 25(OH)D concentrations and the predicted probability of gestational diabetes mellitus (GDM).

**Table 1 nutrients-13-02538-t001:** Participants’ characteristics according to gestational diabetes mellitus (GDM).

Characteristics	Entire Sample (*n* = 1497)		Non-GDM (1453, 97.1%)		GDM (44, 2.9%)		*p*
	*n*	%	*n*	%	*n*	%	
**Socio-demographic characteristics**							
Age (mean ± SD)	32.2 ± 4.9		32.2 ± 4.9		33.8 ± 4.2		0.038
Living area							0.149
North	701	46.8	676	46.6	24	55.8	
Central	392	26.2	379	26.1	13	30.2	
South and East	403	27.0	397	27.3	6	14.0	
Monthly household income ($NT)							0.101
≤59,999	801	53.5	785	54.0	17	39.5	
60,000–99,999	485	32.4	468	32.2	16	37.2	
≥100,000	211	14.1	200	13.8	10	23.3	
Religion							0.001
None	722	48.3	701	48.3	20	46.5	
Buddhism	292	19.5	284	19.6	7	16.3	
Tao	364	24.3	358	24.7	6	14.0	
Other (Yiguandao, Christian, Muslim)	117	7.8	107	7.4	10	23.3	
**Pregnancy-related characteristics**							
Pre-pregnancy BMI (kg/m^2^) (median, IQR)	21.6 (19.8, 24.4)		21.6 (19.8, 24.3)		22.9 (21.1, 26.2)		0.007
Gestational weight gain(kg) (median, IQR)	5.6 (2.0, 9.5)		5.5 (2.0, 9.2)		9.1 (6.2, 11.3)		<0.001
Gestational age							<0.001
1st trimester	418	27.9	414	28.6	2	4.7	
2nd trimester	500	33.4	493	33.9	8	16.3	
3rd trimester	579	38.7	546	37.5	34	79.0	
Gravida							0.112
Primigravida	760	50.8	738	50.8	23	52.3	
2 pregnancies	476	31.8	467	32.1	9	20.5	
≥3 pregnancies	261	17.4	249	17.1	12	27.3	
Parity							0.295
1	916	61.2	892	61.4	23	53.5	
≥2	581	38.8	561	38.6	20	46.5	
Self-reported history of diabetes							0.007
No	1482	99.1	1441	99.2	41	93.2	
Yes	14	0.9	11	0.8	3	6.8	
**Dietary intakes**							
Fat (%) (mean ± SD)	35.9 ± 9.5		35.8 ± 9.5		37.7 ± 8.9		0.214
Protein (%) (mean ± SD)	15.2 ± 3.6		15.2 ± 3.6		16.3 ± 4.4		0.037
Carbohydrate (%) (mean ± SD)	49.7 ±9.9		49.8 ± 9.9		46.1 ± 9.3		0.017
Vitamin D content (median, IQR)	2.8 (1.4, 7.7)		2.8 (1.4, 7.8)		1.7 (0.3, 4.2)		0.055
**Vitamin supplements**							0.012
No relevant supplements	757	50.6	743	51.1	14	31.8	
Vitamin D and/or Calcium	740	49.4	710	48.9	30	68.2	
25(OH)D concentrations (mean ± SD)	25.5 ± 8.9		25.5 ± 8.9		26.6 ± 8.5		0.426

Abbreviations: $NT, New Taiwan dollar; SD, standard deviation; BMI, body mass index; IQR, interquartile range.

**Table 2 nutrients-13-02538-t002:** Univariate logistic regression analysis to estimate odds ratio (and 95% CI) of having gestational diabetes mellitus according to characteristics of pregnant women.

Characteristics	OR	95% CI	*p*
Age (mean ± SD)	1.07	1.01–1.14	0.038
Living area			
North	1.00		
Central	0.96	0.49–1.91	0.918
South and East	0.42	0.17–1.04	0.060
Monthly household income ($NT)			
≤59,999	1.00		
60,000–99,999	1.62	0.81–3.22	0.172
≥100,000	2.32	1.04–5.14	0.039
Religion			
None	1.00		
Buddhism	0.90	0.38–2.13	0.812
Tao	0.58	0.23–1.47	0.252
Other (Yiguandao, Christian, Muslim)	3.23	1.47–7.10	0.004
**Pregnant-related characteristics**			
Pre-pregnancy BMI (kg/m^2^)	1.09	1.02–1.17	0.007
Gestational weight gain(kg)	1.09	1.03–1.15	0.002
Gestational age			
≤2nd trimester	1.00		
3rd trimester	5.89	2.85–12.17	<0.001
Gravida			
Primigravida	1.00		
2 pregnancies	0.59	0.27–1.32	0.202
≥3 pregnancies	1.49	0.72–3.06	0.279
Parity			
1	1.00		
≥2	1.35	0.74–2.48	0.328
**Dietary intakes**			
Fat (%)	1.02	0.98–1.05	0.213
Protein (%)	1.08	1.01–1.17	0.037
Carbohydrate (%)	0.96	0.94–0.99	0.017
Vitamin D content	0.97	0.94–1.01	0.094
**Vitamin D and/or calcium supplement**			
No	1.00		
Yes	2.27	1.18–4.32	0.013

Abbreviations: SD, standard deviation; BMI, body mass index; OR, odds ratio; CI, confidence interval; $NT, New Taiwan dollar.

**Table 3 nutrients-13-02538-t003:** Association between vitamin D during pregnancy and gestational diabetes mellitus.

	Crude Model			Model 1			Model 2		
Vitamin D Levels	OR	95% CI	*p*	AOR	95% CI	*p*	AOR	95% CI	*p*
25(OH)D concentration	1.01	0.98–1.05	0.426	0.97	0.94–1.01	0.174	0.97	0.93–1.01	0.144
Vitamin D insufficiency									
≥32 ng/mL	1.00			1.00			1.00		
<32 ng/mL	0.79	0.39–1.59	0.522	1.34	0.63–2.83	0.449	1.20	0.54–2.69	0.655
Vitamin D deficiency									
≥20 ng/mL	1.00			1.00			1.00		
<20 ng/mL (VDD)	0.94	0.48–1.82	0.845	1.87	0.89–3.92	0.097	2.26	1.03–4.95	0.041

Abbreviations: VDD, vitamin D deficiency; OR, odds ratio; CI, confidence interval; AOR, adjusted odds ratio. Model 1: adjusted for maternal age, household income, religion, pre-pregnancy BMI, gestational weight gain, gestational age, %protein, %carbohydrate, vitamin D, and/or calcium supplement. Model 2: further adjusted for sun exposure, season of blood draw.

## Data Availability

Not applicable.

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
