# Peer review of "Plasma 25(OH)D Concentrations and Gestational Diabetes Mellitus among Pregnant Women in Taiwan"

_nutrients, 2021, doi:10.3390/nu13082538_

Round 1

Reviewer 1 Report

In this study, Pham and colleagues explored the association between maternal plasma vit. D levels and risk of developing gestational diabetes mellitus. This an interesting field of study, despite of whole variability in metabolic and nutritional traits among individuals, which are not often easy to control, and for this reason the contradictory results are numerous. 

In this study, despite it is interesting to address similar goals into different populations, I am concerned with the following aspects:

  • first, this is a prospective or retrospective study? how were patients recruited from the period between June 2017 to February 2019? All patients were enrolled in this period?
  • exclusion criteria were not included. please add
  • the authors considered the impact of sunlight exposure and supplements intake in plasma vit D levels. but, how can you measure the total sunlight exposure? will it be similar with and without sun cream protection? in addition, the forms of vit D used in supplements were considered? what brands patients received? were they biologically active and bioavailable?
  • was this study done according to the Helsinki Declaration?
  • how was sample size calculation done? From 1497 patients, how can you ensure that the difference between 44 GDM and 1453 non-GDM is of statistical robustness? what program was used to ensure such statistical significance? a priori tests were performed?
  • l. 257: in this study what was the average of vit D taken by pregnant women. was it assessed?
  • l. 279-281: were these supplements taken without medical supervision? most patients took vit. D supplements; so, assessing doses, brands and bioavailability of vit D forms ingested is of extreme usefulness and importance to obtain solid conclusions
  • l. 288: control > controlled

Author Response

Dear reviewer 1,

I would like to thank you for your comments and interest in our manuscript.

Please see the attachment for our responses.

Kindly regard,

Thu T.M Pham

Reviewer 2 Report

Line 29, 'VVD' should be described in full name.

Lines 78-83, please specidy the number of population of which the sample has been selected.

The sampling process should be depicted in a table to show readers its representativness. 

Lines 101-102, should be stated as the limitation for your study. 

Ethical considerations should be brought under a separate subheading and all details should be mentioned there.

Line 106, the instrument should be introduced, its contain, domains, questions, scoring system, reliability and validity checkings should be all described.

In the discussion, where you compare your findings with those of other studies, please mention the name of that country. 

Author Response

Dear review 2,

Thank you very much for your comments and interest in our manuscript.

Please see the attachment for our responses.

Kindly regard,

Thu T.M Pham

Round 2

Reviewer 1 Report

Most comments raised were properly addressed. A clear mention to the Helsinki Declaration should be done

Author Response

Dear reviewer 1,

Thank you for your comment and I would like to respond to yours as below.

First, we have stated as "...Institutional Review Board Statement: The study was conducted in accordance with the guidelines of the Declaration of Helsinki and approved by the Institutional Review Board of the government and selected hospitals in Taiwan (IRB number: N201707039).

Informed Consent Statement: Informed consent was obtained from all participants involved in the study.” in lines 312-316.

Second, we have mentioned in section “2.3. Ethical consideration”  as “This study was funded by the Health Promotion Administration, Ministry of Health and Welfare in Taiwan (C1050912), and approved by the Institutional Review Board of the government and selected hospitals (IRB number: N201707039).” in lines 157-160, and “…All participants were asked to obtain informed consent. …” in lines 102-103.

Thank you!

Best regards,

Thu